# Effect of Pectin on the Quality Attributes and Phenolic Composition of Blackberry Jam from Wild and Cultivated Fruits at Different Altitudes

**DOI:** 10.3390/foods14193420

**Published:** 2025-10-03

**Authors:** Adis Veliu, Xhabir Abdullahi, Erhan Sulejmani, Omer Faruk Celik, Mehmet Ali Olcer, Burhan Ozturk

**Affiliations:** 1Department of Food Technology, Faculty of Food and Nutrition, University of Tetova, 1200 Tetovo, North Macedonia; a.veliu222398@unite.edu.mk (A.V.);; 2Department of Food Engineering, Faculty of Agriculture, Ordu University, Ordu 52200, Türkiye; 3Department of Horticulture, Faculty of Agriculture, Ordu University, Ordu 52200, Türkiye

**Keywords:** pectin concentration, altitudinal variation, blackberry jam, sensory evaluation

## Abstract

This study investigated the influence of different pectin concentrations (0%, 0.1%, and 0.5%) on the physicochemical, antioxidant, and sensory properties of blackberry jam (*Rubus fruticosus* L.) prepared from fruits harvested at three altitudinal locations (wild: 998 m; cultivated: 500 m and 1090 m). The jams were analyzed for phenolic profile, antioxidant capacity, color, texture, and sensory attributes. The results showed that altitude strongly affected the phenolic profile and antioxidant capacity, with wild blackberries exhibiting the highest levels of total phenolics, flavonoids, and anthocyanins. Pectin addition in moderate levels (0.1%) enhanced sensory acceptance, particularly in jams from higher altitudes. Furthermore, jams with added pectin showed improved vitamin C retention and reduced bitterness associated with phenolic compounds. Overall, the findings highlight the dual role of pectin in modulating the functional and sensory qualities of blackberry jam, while also demonstrating the impact of altitudinal variation on fruit-derived products.

## 1. Introduction

Blackberries (*Rubus fruticosus* L.) are widely valued for their nutritional quality and economic importance. Globally, more than 400 varieties have been developed, with nearly 100 under extensive cultivation. They are consumed both fresh and processed into products such as juices, jams, purees, concentrates, and desserts, which enhances their availability and commercial potential [1,2]. Blackberries are a rich source of carbohydrates, minerals (notably calcium), vitamins (C, A), and bioactive compounds, including flavonoids, anthocyanins, ellagic acid, and other polyphenols, which are associated with multiple health benefits [3,4,5,6,7,8].

Processing fruits into jams and related products extends shelf life, reduces postharvest losses, and provides consumers with functional foods of high nutritional value [5,9,10]. During jam production, however, heat processing may reduce sensitive compounds such as anthocyanins, which are important contributors to color and antioxidant activity [11]. Pectins are hydrocolloids widely applied in intermediate-moisture foods such as jellies, jams, marmalades, confectionery, pastries, and yogurts because of their gelling and emulsifying properties [12]. They also serve as stabilizers in fruit juices and acidified milk drinks and as fat replacers in ice creams and spreads. The gelling ability of pectin, extensively investigated in relation to structure–function behavior, is promoted by intrinsic factors such as galacturonic acid content, molecular weight, and degree of esterification, and influenced by extrinsic factors including concentration, temperature, pH, soluble solids, and divalent cations [13]. In processing, pectin is typically added late to reduce heat exposure and ensure complete dissolution, with high-methoxyl pectin commonly employed as the gelling and texturizing agent in jams and jellies [14].

Its content varies among fruits, from about 0.26% in strawberries to 0.77% in cherries [15,16,17]. The incorporation of pectin not only influences texture but can also affect the retention of nutritional and sensory qualities. Moreover, jam production provides income opportunities for small-scale producers and contributes to agritourism and rural economic development [18,19,20,21].

Despite these advantages, limited research has focused on wild blackberry varieties that grow abundantly in mountainous regions of the Western Balkans. These wild types are distinguished by their higher acidity and unique flavor, but they remain underutilized compared to cultivated varieties. Exploring their potential in value-added products such as jams is of both scientific and economic interest, particularly given the increasing demand for organic and functional foods in regional and international markets [22].

So, the aim of this study was to determine the effect of different pectin concentrations on the phenolic profile, antioxidant activity, color, texture, and sensory properties of blackberry jams prepared from fruits harvested at different altitudes.

## 2. Materials and Methods

### 2.1. Plant Materials

Blackberry fruit (*Rubus fruticosus* L.) was hand-harvested during the first and second decades of August 2022 from three distinct locations in the northwestern part of the Republic of North Macedonia, categorized by altitude as follows:—F1: Variety from the mountainous area of Poroj village, at an altitude of 998 m (42°02′34.8″ N, 20°58′43.1″ E)—F2: Variety from the city of Tetovo, at an altitude of 500 m (42°00′02.3″ N, 20°58′06.0″ E)—F3: Variety from the mountainous area of Shipkovica village, at an altitude of 1090 m (42°02′08.1″ N, 20°54′55.9″ E). After harvesting, the samples were promptly packed and stored at −20 °C until analyses and jam preparation.

### 2.2. Jam Formulation and Preparation Procedures

The ingredients and preparation method for the nine blackberry jams are outlined in Table 1. The production methodology is adapted from a recipe by Bam Dooel, a company based in Tetovo, North Macedonia. For each batch of jam, the following ingredients are used: 100 g of blackberries, 60 g of sugar (adjust according to desired sweetness), 1 g of citric acid, and between 0.1% and 0.5% of pectin. The preparation begins with cleaning and pressing the blackberries, followed by heating the mixture, and finally, pouring it into jars. The citric acid used in this recipe is sourced from “Maks” in Strumica, North Macedonia, while the pectin is imported from “Danisco” in the Czech Republic.

### 2.3. Physicochemical Analysis

The pH was determined using a portable pH meter (SevenGo Duo™ SG23, Mettler Toledo, Greifensee, Switzerland). To measure titratable acidity and vitamin C, juice was extracted using an electric juicer (Juicer, Philips, Istanbul, Türkiye). For titratable acidity, 10 mL of juice was diluted with 10 mL of distilled water and titrated to a pH of 8.2 using 0.1 mol/L sodium hydroxide, with results expressed as grams of malic acid per 100 g of juice. For vitamin C, 0.5 mL of juice was combined with 0.5% oxalic acid to make a total volume of 5 mL. An ascorbic acid test strip (Merck, Darmstadt, Hessen, Germany) was then dipped into the mixture for 2 s, and the results were reported as mg per 100 g of juice [23].

### 2.4. Individual Phenolic Compounds

Approximately 40–50 mL of the resulting homogenate was placed in a Falcon tube and stored at 4 °C until the bioactive analyses were conducted. The individual phenolics in the samples were detected using high-performance liquid chromatography (HPLC), following a modified method based on Ozturk et al. [23]. For the analysis, a Thermo Scientific Ultimate 3000 UHPLC system, a Hypersil GD phenyl column (Thermo Scientific, Waltham, MA, USA), and a UV detector (Model number: DAD-3000, USA) were used, operating at a wavelength of 230 nm. The jam samples were prepared by mixing them with distilled water in a 3:1 ratio, followed by centrifugation at 15,000× *g* for 10 min. The resulting supernatant was filtered through a 0.45 µm pore size filter before being loaded onto the UHPLC system. The analytes were separated using a Hypersil GD column (250 × 3.0 mm, 5 µm) maintained at 35 °C. The mobile phase consisted of 0.028% aqueous sulfuric acid (solvent A) and 100% acetonitrile (solvent B), with compounds detected at 230 nm. The total run time for the analysis was 60 min, and the injection volume for each sample was set at 20 µL, with the mobile phase flow rate maintained at 1.0 mL/min. The results are reported in milligrams per kilogram of fresh weight (mg/kg FW).

### 2.5. Total Phenolics, Total Flavonoids and Total Monomeric Anthocyanin

Total phenolic compounds were measured using Folin–Ciocalteu’s reagent [23]. The procedure involved mixing the fruit extract with distilled water, reagent, and sodium carbonate, and then incubating the mixture for 2 h. The absorbance was measured at 760 nm, and the results were expressed as grams of gallic acid equivalents per kilogram of fresh weight. For the total flavonoid content [24], fruit extracts were combined with methanol, aluminum nitrate, and ammonium acetate, and then incubated in the dark for 40 min. The absorbance reading was taken at 415 nm, with results reported as grams of quercetin equivalents (QE) per kilogram. The anthocyanin content was measured using the pH difference method, which involved measuring the absorbance at 700 nm at two different pH levels (1.0 and 4.5) [25]. The results were reported in grams of cyanidin-3-glucoside (C3GE) per kilogram of fresh weight.

### 2.6. DPPH and FRAP Antioxidant Activity

The antioxidant activity of the sample extract was determined using the DPPH and FRAP methods. For the DPPH method, 500 µL of the extract was mixed with 2.5 mL of ethanol and 0.1 mM DPPH solution. After allowing it to incubate in the dark, the absorbance was measured at 517 nm. The results were expressed as millimoles of Trolox equivalents (TE) per kilogram of fresh fruit. In the FRAP method, 1.15 mL of phosphate buffer was combined with 100 µL of the extract and 1.25 mL of potassium ferricyanide. After incubation and cooling, trichloroacetic acid and ferric chloride were added, and the absorbance was measured at 700 nm. The results were expressed as millimoles of Trolox equivalents (TE) per kg FW.

### 2.7. Color Characteristics

The color values (L*, a*, and b*) of the jams were measured using a portable color meter (Konica-Minolta, CR-400, Tokyo, Japan). Because the samples were dark, they were initially diluted at a 1:1 ratio with distilled water, then vortexed, shaken, and centrifuged to collect the supernatant. This supernatant was further diluted at a ratio of 1:4 for measurement, resulting in a final dilution of 1:10. Four milliliters of the diluted sample were placed in a 5.0 mL vial, and the color was measured with a colorimeter, with at least three readings taken to calculate average values. The browning index (BI) was calculated using the following equation [26]:BI = [100 × (x − 0.31)]/0.172
where x isX = (a* + 1.75 L*)/(5.645 L* + a* − 3.012 b*)

### 2.8. Texture Characteristics

For texture analysis, 30 g of each jam sample were placed in 40 mL glass jars (50 mm in height and 30 mm in diameter). A texture analyzer (TA-XT2, Stable Micro Systems, Godalming, Surrey, UK) equipped with a 25 mm probe and a 5.0 kg load cell was utilized to perform measurements at a temperature of 25 ± 2 °C. The pre-test speed was set to 3 mm/s, the test speed to 1.0 mm/s, and the post-test speed to 2.0 mm/s, with a test distance of 15 mm and a trigger force of 5.0 g. The parameters recorded included rupture strength, gel strength, brittleness, adhesiveness, and consistency [10].

### 2.9. Sensory Analysis

To ensure a reliable sensory evaluation, twelve non-smoking participants (eight men and four women) aged 24–54 from the Department of Horticulture, Faculty of Agriculture, Ordu University, were recruited. Before the evaluation, participants received detailed information about the study and a brief orientation to standardize scoring procedures (using a 5-point hedonic scale). The attributes assessed included appearance, color, odor, texture, taste, and overall consistency. Each sample was presented in randomly numbered plastic containers to reduce potential bias. Participants consumed a cracker before assessing the next sample and rinsed their mouths with water to cleanse the palate. Although the trained panel provided useful insights into the sensory characteristics, the results should be interpreted as preliminary and not as definitive evidence of consumer acceptability.

### 2.10. Statistical Analysis

Data from three localities (samples F1, F2, and F3) were included in a randomized design consisting of nine samples and two independent replicates (trials). The response variables analyzed included composition, color, and phenolic compounds. A general linear model was used to conduct ANOVA, treating locality effects as fixed terms and replicates as random terms. Significance was assessed using the Duncan test at a confidence level of *p* < 0.05. To investigate relationships among phenolics within the localities, principal component analysis (PCA) based on the correlation matrix and heat mapping were performed using SPSS (IBM Corporation, version 21.0, Armonk, NY, USA) and XLSTAT software (Lumivero XLSTAT statistical software, version 2025.1).

## 3. Results and Discussion

### 3.1. Physicochemical Properties and Color Profile

Table 2 presents the physicochemical properties of fresh blackberry fruits and the corresponding jams. Mikulic et al. [27] found no significant differences in the pH levels of juices among different cultivars. They noted that blackberry juice generally has a relatively low pH compared to other fruits, ranging from 2.7 to 3.1. The pH of fruit jams increased with the addition of pectin, likely due to its buffering effect on acidity.

Similarly, Siddiqui et al. [28] examined the effects of varying pectin concentrations on the physicochemical and sensory qualities of jams and observed that higher pectin concentrations led to increased pH values. According to Ramadhan and Trilaksani [29], the ideal pH range for gel formation in jam is between 3.0 and 3.4. While there was no significant difference in titratable acidity among the samples, F2 (1.12) and F3 (1.06) exhibited higher acidity levels than F1 (0.73). Reported pH values for jams vary by fruit: strawberry (2.91–3.47), raspberry (2.98–3.04), sour cherry (3.28–3.47), white cherry (3.57), and apricot (2.09–4.15) [30,31,32,33]. In this study, the pH of blackberry jams ranged from 3.08 to 3.52, which aligns closely with strawberry and sour cherry, slightly exceeds raspberry, and is generally lower than that of apricot. According to Uribe-Wandurraga et al. [34], a pH value below 4.0 indicates microbiological stability in jams. Typically, fruit jams have a pH between 2.8 and 3.5, which is necessary for proper pectin gelation [35]. Additionally, while additives like pectin can influence the physical and chemical characteristics of the final product in various ways [36], acidity levels also varied across different samples and formulations. The highest acidity was noted in J-F3-0P (1.32 g per 100 g of malic acid). However, the effect of pectin on acidity was dependent on the specific formulation. The acidity of F1 jams increased with the addition of pectin, indicating a possible interaction between the acidic components and pectin. In contrast, F2 jams showed a decrease in acidity, while the acidity of F3 jams remained stable. These variations can likely be attributed to the different fruit compositions and buffering characteristics of each formulation. Vitamin C content was significantly affected by the formulation and the presence of pectin. The highest levels of retained vitamin C were found in F1 jams with 0.1% and 0.5% pectin, measuring 175.33 mg and 173.33 mg per 100 g, respectively. This suggests that pectin may have played a protective role against oxidative degradation. On the other hand, F2 jams consistently had the lowest vitamin C content, particularly in the absence of pectin, which measured only 69.67 mg per 100 g. The trends across the formulations imply that pectin not only influences texture and moisture retention but may also help stabilize sensitive nutrients like vitamin C during processing and storage, especially in the F1 and F3 samples. These findings highlight the multifaceted role of pectin in jam formulation, affecting both physical structure and chemical stability, as well as the nutritional quality of the final product. Jams prepared from F3 exhibited vitamin C levels ranging from 133.33 to 175.33 mg per 100 g, comparable to or exceeding those reported by Noila et al. [1] for fresh ‘Karaka Black’ and ‘Thornfree’ blackberries, indicating that the nutrient remains significantly retained even after processing.

Color analysis of blackberry jam samples, prepared without pectin and with different concentrations of 0.1% and 0.5%, revealed significant differences related to the sample composition (Table 3). The color attributes varied significantly depending on the fruit source and the amount of additive used. The lightness (*L**) values ranged from approximately 25.24 to 26.90, indicating generally dark products typical of blackberry jams. The sample J-F2-0P exhibited the highest lightness, suggesting a slightly lighter color compared to other formulations, possibly due to differences in fruit characteristics or processing conditions. Conversely, J-F1-0P had the lowest lightness value, indicating a darker appearance.

The red-green coordinate (*a**) values were relatively low but exhibited notable differences among samples. The highest level of redness was observed in J-F2-0.1P, suggesting better retention of the anthocyanin pigments responsible for its red hue. In contrast, J-F3-0.1P displayed the lowest redness, which could indicate either pigment degradation or a lower concentration of pigments in that sample. Similarly, the yellow-blue coordinate (*b**) varied among samples, with J-F3-0.1P showing the highest yellowness and J-F2-0.1P the lowest. This indicates differences in pigment composition or processing effects. The browning index (BI) also varied significantly; J-F3-0P had the highest BI value, indicating more intense non-enzymatic browning reactions, which may negatively impact the product’s sensory quality and shelf life. Lower BI values in the J-F1-0P and J-F3-0.1P samples suggest less browning and potentially better color stability. Moreover, the brown color index was more pronounced in samples J-F3-0P and J-F2-0.5P, indicating that higher amounts of pectin contributed to the deepening of the brown color through oxidation during processing. Higher-altitude varieties (F1 and F3) may produce jams with a deeper color but also more browning due to a higher presence of phenolics. In contrast, the lower-altitude variety (F2) produces lighter jams with potentially better pigment stability and less browning. The inclusion of pectin in blackberry jam has enhanced the color parameters; a concentration of 0.1% improved the red color, while a higher concentration of 0.5% emphasized the brown color of the jam. Samples without pectin resulted in less optimal color quality. Therefore, moderate pectin addition (around 0.1%) is beneficial for color preservation in blackberry jam, effectively balancing pigment protection and browning control. The color parameters of fresh blackberries, as reported by Mikulic-Petkovsek et al. [27], indicate a dark purple appearance, with *L** values ranging from 19.8 to 23.9. In comparison, the jams produced by J-F appear lighter, with *L** values between 25 and 27. In contrast, the jams described by Gurel and Velioglu [37] are significantly darker, with *L** values ranging from 1.2 to 5.36. This suggests that the J-F formulations retain more lightness, likely due to milder processing methods or lower sugar levels, while the darker jams may reflect higher sugar content, more intense heating, or concentration effects. Redness (a*) is better preserved in the J-F jams. In contrast, some of the jams from Gurel and Velioglu [37] exhibit slight green shifts as a result of processing. Yellowness (*b**) remains minimal across all jams but is somewhat more pronounced in the samples from the literature, likely due to Maillard reactions or caramelization.

### 3.2. Results of Total Phenolics, Flavonoids, Anthocyanins and Antioxidant Assays

The samples without pectin demonstrate a relatively higher content of total phenolic compounds, indicating that the addition of pectin to blackberry jam reduces the overall amount of these compounds (Table 4).

The F1 variety contains a richer concentration of phenolic compounds compared to the F2 variety, which has a lower level of total phenolic compounds. The results for the total amounts of phenolics and flavonoids indicate that their values range from 1.08 to 1.77 g GAE per kg and from 0.19 to 0.87 g QE per kg, respectively. In general, jam made from berries of the F1 variety has shown a higher total flavonoid content, particularly in samples without added pectin. In contrast, samples from the F2 variety exhibited lower flavonoid concentrations, although these levels gradually increased after the addition of pectin. Samples from the F3 variety, however, displayed stable total flavonoid amounts across all three levels of pectin used. These results suggest that the effect of pectin on total flavonoid content is influenced by the geographical origin of the fruit and its initial composition, which may indicate varying biochemical interactions during the jam preparation process [3]. When assessing antioxidant activity using the FRAP method (Ferric Reducing Antioxidant Power), samples from the F1 variety demonstrated the highest antioxidant activity, with a significant improvement noted upon the addition of pectin. For the F2 variety, adding 0.1% of pectin enhanced the antioxidant activity, while in the F3 variety, pectin negatively impacted antioxidant performance. Overall, the influence of pectin on antioxidant capacity appears to be dependent on the blackberry variety, suggesting distinct interactions between bioactive compounds and pectin for each type [14]. The F1 samples provided the most promising results; the blackberry jam made from this variety, when supplemented with pectin, exhibited a significant increase in anthocyanin content. In the F2 variety, the natural anthocyanin levels were lower without pectin, but the inclusion of pectin resulted in an increase in their concentration. For the F3 variety, there was only a slight increase in anthocyanins, and the effect of pectin was insufficient to produce further improvements beyond a certain threshold. Based on the findings of Marjanović [38], J-F1 exhibits higher flavonoid levels compared to standard blackberry, while fresh blackberry contains greater total phenolics and anthocyanin content. This indicates that J-F1 is more suitable for applications requiring high flavonoid content, whereas blackberry is superior for achieving elevated anthocyanin levels or stronger antioxidant activity.

In terms of antioxidant activity measured by the DPPH standard, the F1 samples exhibited the highest activity, which can be attributed to the addition of pectin. The F2 samples showed a positive response to increased pectin levels, while the F3 samples exhibited a decline in antioxidant activity with the addition of pectin. These findings suggest specific interactions between pectin and phenols: in the F3 variety, pectin appears to block or mask the antioxidant properties of phenols. In contrast, in the F1 variety, pectin seems to help stabilize or enhance the availability of these antioxidant properties.

The 3D graph illustrates the relationship between antioxidant activity (measured by the FRAP method), total flavonoids, and total phenolic compounds in various blackberry jam samples (Figure 1). The total phenolic compounds values are displayed on the *X*-axis, while the *Y*-axis ranks the samples based on variety and pectin content. The *Z*-axis represents the amounts of total flavonoids. The surface color indicates the FRAP values (antioxidant activity), revealing that the F1 sample exhibits increased antioxidant properties as the percentage of pectin rises, as shown by the red peak in the upper portion of the graph. Conversely, the F2 variety samples show lower levels of total phenolics and FRAP antioxidant activity, indicated by the blue-green area on the graph. The F3 variety samples present intermediate levels of total phenolic compounds (1.28–1.35 mmol TE kg^−1^) and FRAP (28.46–32.73), which correspond to the yellow region of the graph. The data clearly indicate a positive correlation between FRAP antioxidant activity and the amounts of flavonoids and phenolic compounds, particularly in the F1 jam samples containing pectin. The addition of pectin improves both the stability and antioxidant activity of blackberry jam. It is important to note that the antioxidant activity of these samples is somewhat lower than that of unprocessed fruits reported by Paunović et al. [39]. This reduction is expected, as the heating involved in jam preparation can lead to the loss of certain antioxidant compounds, such as vitamin C and specific polyphenols. Nevertheless, the measured DPPH activity (0.55–0.69 mmol TE 100 g^−1^) is comparable to that of processed products like jam in the aforementioned study, suggesting that some antioxidants are preserved despite the thermal treatment.

### 3.3. Results of Individual Phenolic Compounds

The analysis of individual phenolic compounds in the blackberry jam samples revealed significant differences (*p* < 0.05) influenced by both the variety (J-F1, J-F2, J-F3) and the pectin concentration (0%, 0.1%, 0.5%) (Table 5). The results indicate that the content of specific phenolics varies not only with the type of raw material or processing method but also with the interaction of added pectin, which is a common gelling agent in fruit preserves. Among the three varieties, J-F1 consistently showed higher concentrations of certain phenolic compounds, such as 4-aminobenzoic acid (PABA), epicatechin, and ferulic acid, particularly when higher levels of pectin were added. This suggests that the composition of J-F1, influenced by its fruit origin and processing method, effectively preserves or enhances phenolic compounds during jam production. This pattern is clearly illustrated in the heat map, where yellow indicates higher concentrations of these phenolic compounds, while dark blue represents lower levels (see Figure 2A). In contrast, J-F2 samples exhibited the lowest levels of most phenolics across all pectin concentrations, indicating a weaker phenolic profile. This may be attributed to differences in the raw material, thermal processing, or interactions with pectin, which could reduce the extractability or stability of the phenolics. Meanwhile, J-F3 samples were characterized by high levels of 4-hydroxybenzoic acid and chlorogenic acid, especially with the addition of pectin. This suggests that this formulation either contains these compounds in higher native quantities or is more effective at extracting and stabilizing them during processing. The addition of pectin at varying concentrations (0%, 0.1%, and 0.5%) significantly (*p* < 0.05) influenced the phenolic profile. Specifically, the 0.5% pectin level resulted in notably higher concentrations of several important phenolic compounds, including chlorogenic acid, ferulic acid, and 4-aminobenzoic acid. For example, chlorogenic acid levels reached 29.83 mg/kg in the J-F3-0.1P sample and 28.89 mg/kg in the J-F1-0.5P sample, while ferulic acid peaked at 14.79 mg/kg in the J-F1-0.5P sample. These increases indicate that pectin may enhance the extraction, solubilization, or protection of certain phenolic compounds during thermal processing, likely by forming gels that reduce oxidative degradation.

These results align with the findings of Benedek et al. [40], who identified chlorogenic, neochlorogenic, p-coumaric, and ferulic acids as the main phenolic compounds in blackberry jams. However, the levels of these compounds were quantitatively higher in our study. In contrast, lower levels of hydroxycinnamic acids (isomers of chlorogenic acid) were observed in the ‘Čačanska Bestrna’ blackberry and ‘Willamette’ raspberry [41]. According to Ifie et al. [42], hydroxycinnamic acids contribute to the generation of stable anthocyanin-derived pigments by condensing with anthocyanins, which might help maintain color intensity. Interestingly, some compounds, such as catechin, epicatechin, and caffeic acid, showed a decrease or an inconsistent trend with increasing pectin levels. This could be due to the potential binding of these compounds with pectin or degradation during heating, suggesting that their responses to pectin are compound-specific. 4-Aminobenzoic acid (PABA) exhibited a significant (*p* < 0.05) increase with pectin, particularly in the J-F1 samples, indicating that pectin may enhance the retention or release of this compound from the jam matrix. In contrast, protocatechuic acid remained low across all samples, suggesting that neither the formulation nor the addition of pectin significantly (*p* < 0.05) influences its content. 4-hydroxybenzoic acid was consistently high in the J-F3 samples, regardless of the pectin level, implying that its presence is more dependent on the formulation than on treatment. Chlorogenic acid and ferulic acid were both notably enhanced by pectin, especially in the J-F3 and J-F1 formulations. On the other hand, catechin and epicatechin tended to decrease with higher pectin concentrations, particularly in J-F1, indicating potential instability or reduced extraction in the presence of pectin. The differences in compound levels between samples were statistically significant (*p* < 0.05), as indicated by different superscript letters assigned to the mean values. This reinforces the conclusion that both the type of formulation and the concentration of pectin play crucial roles in determining the phenolic content of blackberry jams. This study highlights the complex interaction between pectin concentration and the retention of phenolic compounds in blackberry jam. While adding pectin (especially at 0.5%) enhances certain compounds like chlorogenic acid, ferulic acid, and PABA, it may suppress others, such as catechin and epicatechin. Furthermore, these effects are strongly influenced by the jam formulation, underscoring the need to optimize both raw material quality and processing parameters collaboratively. The findings discussed are important for food scientists and producers who seek to enhance the nutritional value and antioxidant potential of fruit-based products. Research shows that phenolic antioxidants do not have a consistent reaction order. The variations observed in processed fruits can be attributed to several reactions, including caramelization, the Strecker and Maillard pathways, the degradation of larger phenolic compounds into smaller, antioxidant-active molecules, and transformations of anthocyanins [40].

### 3.4. Results of Textural and Sensory Analysis of Blackberry Jam

The analysis of the textural properties of blackberry jam revealed that gelling strength is highest in the sample J-F1-0P (the sample without pectin) and lowest in J-F2-0.5P (the sample with 0.1% of pectin) (Table 6). This finding underscores the significant impact of the initial composition and the fruit variety’s growing location on gel structure formation, indicating that the addition of pectin does not always enhance gelling strength. The breaking strength was found to be greater in J-F1-0.1P and lower in J-F2-0P. This suggests that a small amount of pectin can increase the structural strength, while larger amounts may not produce a linear effect, likely due to the potential for non-uniform network formation. Creating pectin gels requires a precise balance of pectin, sugar, acid, and water. An excess of pectin can compromise the gel structure, making it too dense or unbalanced. Additionally, fruits like F1, F2, and F3 naturally contain varying amounts of pectin. When extra pectin is introduced, the polymer chains may struggle to link up properly, which can result in gel separation or weakness. Consequently, this leads to a gel that is less smooth and less strong. Research by Yang et al. [43] also indicated that a reduction in pH decreases gel hardness. This occurs because lower pH levels diminish the dissociation of galacturonic acid residues, which restricts electrostatic interactions and the development of the gel network. Brittleness was found to be higher in J-F2-0P and lower in J-F1-0.5P. Here, the presence of pectin and the origin of the fruit influence the gel structure, with lower brittleness values indicating a stronger and more compact gel. The adhesion showed negative values, indicating the tensile force during detachment, with the highest value in J-F1-0P and the lowest in J-F1-0.1P. Higher adhesion values are associated with the gel network structure. Figure 2B presents a heat map that illustrates the intensity of sensory parameter values for each sample. Samples M1 (J-F1-0P) and M3 (J-F3-0P) received lower scores for external appearance and overall acceptability, indicated by red and black shades. In contrast, samples M2 (J-F2-0P), M7 (J-F1-0.5P), and M9 (J-F3-0.5P) are rated higher for aroma and color preference. Meanwhile, samples M4 (J-F1-0.1P), M5 (J-F2-0.1P), and M6 (J-F3-0.1P) exhibit balanced values in consistency and texture.

The consistency of the samples varies, with the highest consistency found in J-F1-0P and the lowest in J-F2-0.1P. This indicates that the samples from the F1 variety, which do not contain added pectin, have a very high consistency due to their natural pectin content and other components present in the fruit. F3 jams display intermediate textural characteristics compared to the other two altitudinal origins. Although adding pectin modestly improves gel firmness and reduces brittleness and adhesiveness, the effect is less pronounced than in F1 samples, likely due to the inherent pectin content and physicochemical properties of the F3 fruit. Pectin is crucial for forming the gel network that binds water and fruit solids, resulting in the semi-solid texture that consumers expect. The appropriate concentration of pectin ensures that the jam is spreadable, cohesive, and stable, which significantly (*p* < 0.05) impacts its sensory acceptance [44].

Table 7 presents the sensory characteristics of the jam preparations. The addition of pectin at varying concentrations (0%, 0.1%, and 0.5%) had a significant (*p* < 0.05) effect on the sensory attributes of the blackberry samples across the three studied cultivars (J-F1, J-F2, J-F3). While the odor of the samples remained mostly unchanged with the addition of pectin-resulting in similar scores for all treatments—there were notable improvements in other sensory attributes.

The color and appearance of the samples improved as the concentration of pectin increased. Samples treated with 0.1% and 0.5% pectin showed significantly (*p* < 0.05) better visual appeal compared to the untreated controls, indicating that pectin may enhance color stability or intensification, thereby making the product more attractive. Additionally, the taste was positively affected by the presence of pectin. Higher concentrations, particularly at 0.5%, led to improved taste scores. This improvement may be attributed to pectin’s effect on mouthfeel and its potential interactions with flavor compounds, which enhance overall flavor perception. Pectin also significantly (*p* < 0.05) improved the consistency and texture of the samples. The treated samples exhibited greater thickness and a more desirable texture, which aligns with pectin’s established role as a gelling and stabilizing agent in food products. These texture improvements likely contributed to the enhanced mouthfeel experienced by the taste panelists. Pectin plays a key role in forming a network and thickening jams during processing; however, excessively high viscosity is undesirable [45]. Overall, the combined positive effects of pectin on color, taste, consistency, and texture resulted in increased overall acceptability of the samples. Generally, samples containing pectin were preferred over those without, with the 0.1% and 0.5% pectin treatments receiving the highest scores for overall acceptability. These findings highlight the beneficial role of pectin as an additive in improving sensory quality and consumer appeal in blackberry-based products. Similarly, studies by Pebriawati et al. [46] and Sharma et al. [47] demonstrated that pectin contributes to achieving optimal gel consistency, which in turn affects color stability, texture integrity, and flavor retention during storage. Consequently, treatments with appropriate pectin levels maintained higher sensory scores for color, texture, and flavor over time. Thus, gelling properties make pectins one of the most widely used plant polysaccharides in food design, with applications in jellies, jams, marmalades, and confectionery products, while beyond their technological role, they also serve as an important source of soluble dietary fiber and contribute significantly to human health [48,49]. Finally, food applications benefit from pectin, a complex polysaccharide whose β-(1,4)-linked D-galacturonic acid backbone and diverse pendant groups determine its functionality [50,51]. Gelation and other functional properties of pectin are influenced by its amphiphilic nature, arising from hydrophilic groups (hydroxyl, carboxyl) and hydrophobic groups (ester, amide, methyl, acetyl) [51,52,53]. Gelation and other functional properties of pectin are influenced by acetyl (CH_3_CO–) and methoxy (CH_3_O–) groups, which affect solubility, molecular interactions, and gel formation [54]. However, a higher degree of acetylation reduces interactions among pectin chains, limiting gel formation [55].

### 3.5. Principal Component Analysis (PCA) Results

The PCA results and heat map data clearly indicate that the texture, phenolic composition, and sensory profile of blackberry jam are significantly (*p* < 0.05) influenced by both the fruit origin and the addition of pectin. Figure 3A presents a biplot generated from the PCA, illustrating the relationships between texture attributes and color parameters. Samples M1 (J-F1-0P) and M3 (J-F3-0P) are noteworthy for their high rupture strength, which is due to the absence of pectin. Typically, pectin reduces gel strength and rupture force while enhancing spreadability and lowering stickiness. These samples, with the exception of M2, are distinctly situated on the positive side of the F1 axis (46.00%) in the PCA plot. In contrast, jams made from the F2 variety are clearly grouped on the negative side of the F1 axis, indicating their weaker texture regardless of pectin addition.

Samples M1 (J-F1-0P), M4 (J-F1-0.1P), and M7 (J-F1-0.5P) are notable for their high content of phenolic compounds, which is likely due to their specific interaction with the F1 variety. In the PCA plot (Figure 3B), these samples are clearly separated from the positive side of the F2 axis (29.30%), while samples from the F2 and F3 varieties are positioned on the negative side. Sample M5 (F2 with 0.1% pectin) exhibits a more moderate phenolic acid profile. Additionally, the F3 variety combined with 0.5% of pectin contains a significantly (*p* < 0.05) high level of 4-hydroxybenzoic acid. Interestingly, the F1 samples without pectin show greater variation and a higher concentration of specific compounds, suggesting that the addition of pectin might reduce certain phenolic compounds.

Figure 3C presents a biplot derived from PCA, illustrating the relationships among sensory variables such as jam color, taste, aroma, texture, consistency, appearance, and overall acceptability (represented by red arrows). The horizontal axis F2 (11.88%) and the vertical axis F3 (4.43%) together explain 16.32% of the total variability, which is within the expected range for multivariate sensory analysis. The PCA results indicate that taste, texture, and consistency are the main factors influencing the overall acceptability of blackberry jam [46,47].

The addition of 0.1% of pectin (in samples M4, M5, and M6) positively impacts the jam’s formation without adversely affecting the sensory quality. These samples are grouped on the negative side of the F1 axis and the positive side of the F2 axis. However, a pectin concentration of 0.5%, particularly in samples M7 and M9, negatively impacts aroma and acceptability [37].

Figure 4 illustrates a dendrogram that shows the hierarchical clustering of samples based on their sensory characteristics. At a dissimilarity level of approximately 25, two main groups, C1 and C2, were identified. Group C1 consists of samples M1 (J-F1-0P) and M3 (J-F3-0P), which are similar to each other but distinctly different from the other samples. This indicates that M1 and M3 possess unique sensory attributes compared to the rest. The C2 group includes the remaining samples, forming a homogeneous cluster with minimal differences among them.

## 4. Conclusions

This study demonstrated that altitude and pectin concentration significantly affect the phenolic composition, antioxidant capacity, texture, and sensory properties of blackberry jams. Higher-altitude fruits (F1, 998 m) develop richer phenolic profiles, which help preserve vitamin C during processing but can lead to stronger astringency and firmer textures. Pectin addition mitigates these effects, improving spreadability, mouthfeel, and overall sensory acceptability.

The findings highlight that controlled use of pectin can enhance both the nutritional and sensory qualities of jams from wild and cultivated blackberry varieties. These results have practical implications for small-scale producers and the food industry, suggesting that adjusting pectin levels according to fruit origin and composition can optimize product quality, consumer satisfaction, and market potential, especially for wild high-altitude berries with unique flavor profiles.

## Figures and Tables

**Figure 1 foods-14-03420-f001:**
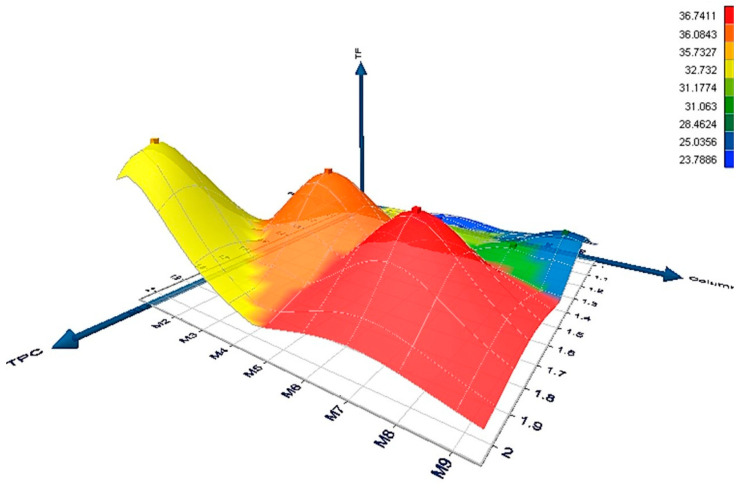
The 3D graph of the correlation between antioxidant activity, total flavonoids and total phenolics in blackberry jam samples.

**Figure 2 foods-14-03420-f002:**
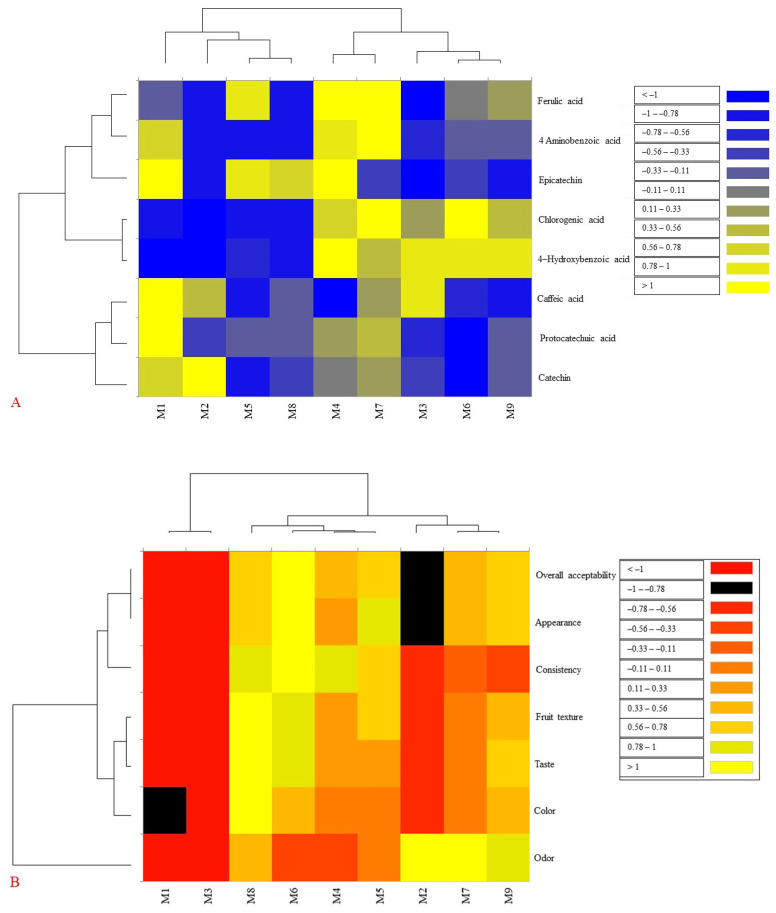
Visualization of clustering patterns of jam samples and variables based on individual phenolic compounds (**A**) and sensory characteristics (**B**) using heat map.

**Figure 3 foods-14-03420-f003:**
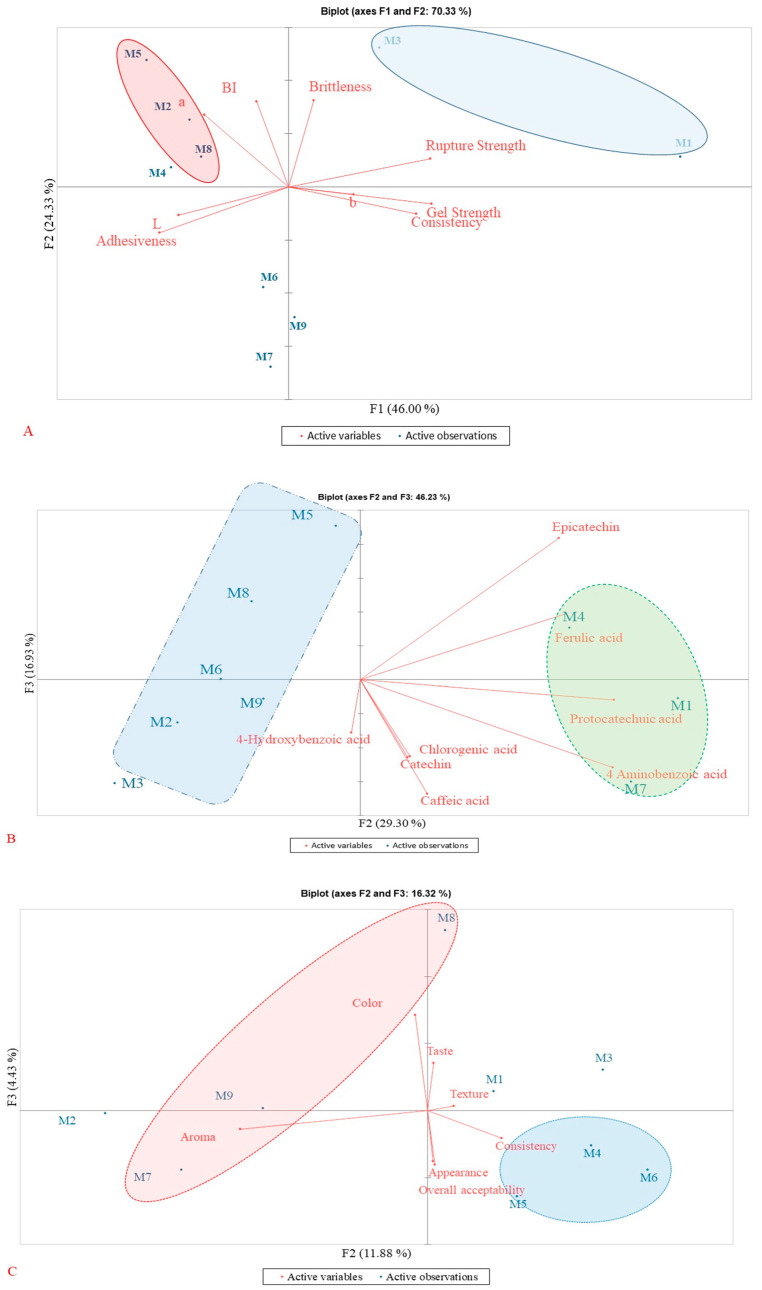
Biplot graph based on texture (**A**), individual phenolic compounds (**B**) and sensory analysis (**C**) jam samples.

**Figure 4 foods-14-03420-f004:**
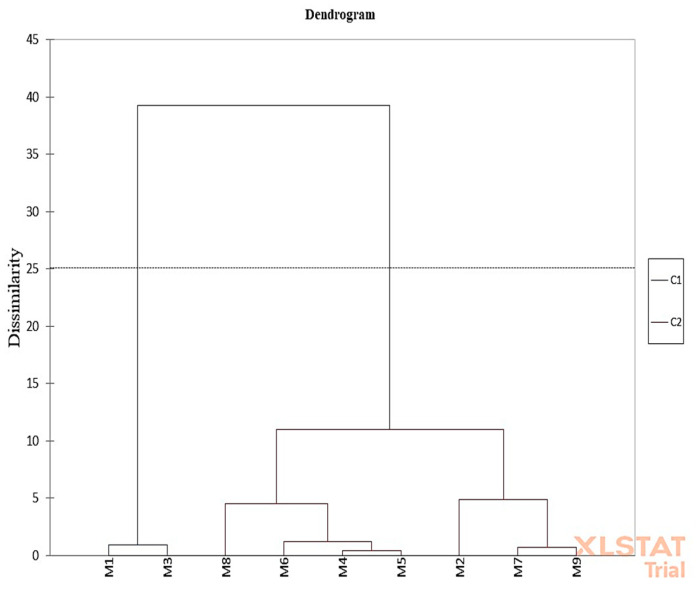
Dendrogram shows hierarchical clustering of samples according to their sensory properties.

**Table 1 foods-14-03420-t001:** Composition of ingredients (%) in making blackberry jams.

Jam Samples	Code	Blackberries	Sugar	Citric Acid	Pectin
J-F1-0P J-F2-0P J-F3-0P	M1 M2 M3	62	37	1	-
J-F1-0.1P J-F2-0.1P J-F3-0.1P	M4 M5 M6	62	37	0.9	0.1
J-F1-0.5P J-F2-0.5P J-F3-0.5P	M7 M8 M9	62	37	0.5	0.5

Blackberry variety fruit locations: F1 (998 m), F2 (500 m), F3 (1090 m).

**Table 2 foods-14-03420-t002:** Physicochemical properties of fresh blackberry fruits and jams.

Fruit Samples	pH	Titratable Acidity (g Malic Acid 100 g^−1^)	Vitamin C (mg 100 g^−1^)
F1	2.90 ± 0.02 ^a^	0.73 ± 0.08 ^a^	239.1 ± 0.20 ^a^
F2	2.89 ± 0.01 ^a^	1.12 ± 0.25 ^a^	266.4 ± 2.95 ^a^
F3	3.09 ± 0.01 ^b^	1.06 ± 0.03 ^a^	449.5 ± 2.44 ^b^
Jam samples			
J-F1-0P	3.52 ± 0.02 ^f^	1.01 ± 0.00 ^b^	162.33 ± 3.21 ^f^
J-F2-0P	3.08 ± 0.03 ^a^	1.01 ± 0.00 ^b^	69.67 ± 5.03 ^a^
J-F3-0P	3.28 ± 0.01 ^c^	1.32 ± 0.02 ^f^	125.33 ± 1.53 ^d^
J-F1-0.1P	3.40 ± 0.02 ^e^	1.10 ± 0.01 ^d^	175.33 ± 5.03 ^g^
J-F2-0.1P	3.18 ± 0.02 ^b^	0.85 ± 0.00 ^a^	88.67 ± 0.58 ^b^
J-F3-0.1P	3.37 ± 0.01 ^d^	1.02 ± 0.01 ^b^	124.67 ± 1.53 ^d^
J-F1-0.5P	3.39 ± 0.01d ^e^	1.16 ± 0.00 ^e^	173.33 ± 1.53 ^g^
J-F2-0.5P	3.21 ± 0.02 ^b^	1.09 ± 0.01 ^c^	99.67 ± 6.81 ^c^
J-F3-0.5P	3.40 ± 0.02 ^e^	1.11 ± 0.01 ^d^	133.33 ± 1.15 ^e^

Values are expressed as mean ± standard deviation. ^a–g^ Values marked with different letters in the same column are significantly different (*p <* 0.05), as determined by Duncan’s multiple range test.

**Table 3 foods-14-03420-t003:** Color characteristics of blackberry jam samples.

Samples	*L**	*a**	*b**	BI
J-F1-0P	25.24 ± 0.21 ^a^	0.43 ± 0.17 ^a^	0.41 ± 0.07 ^b^	2.78 ± 0.19 ^ab^
J-F2-0P	26.90 ± 0.47 ^d^	0.85 ± 0.10 ^c^	0.23 ± 0.03 ^ab^	3.07 ± 0.33 ^ab^
J-F3-0P	25.76 ± 0.39 ^ab^	0.83 ± 0.13 ^d^	0.47 ± 0.06 ^c^	4.08 ± 0.63 ^c^
J-F1-0.1P	26.29 ± 0.24 ^bc^	0.87 ± 0.07 ^d^	0.32 ± 0.18 ^bc^	3.50 ± 0.61 ^bc^
J-F2-0.1P	25.87 ± 0.45 ^bc^	1.09 ± 0.08 ^e^	0.10 ± 0.08 ^a^	3.35 ± 0.15 ^a–c^
J-F3-0.1P	26.32 ± 0.25 ^bc^	0.32 ± 0.12 ^a^	0.50 ± 0.07 ^c^	2.72 ± 0.09 ^ab^
J-F1-0.5P	26.42 ± 0.40 ^cd^	0.58 ± 0.01 ^bc^	0.27 ± 0.10 ^a–c^	2.54 ± 0.29 ^a^
J-F2-0.5P	25.99 ± 0.26 ^bc^	0.68 ± 0.07 ^cd^	0.46 ± 0.20 ^c^	3.61 ± 0.87 ^bc^
J-F3-0.5P	25.99 ± 0.18 ^bc^	0.58 ± 0.11 ^a–c^	0.28 ± 0.07 ^a–c^	2.61 ± 0.26 ^a^

Values are expressed as mean ± standard deviation. ^a–e^ Values with different letters in the same. column are significantly different (*p* < 0.05). as determined by Duncan’s multiple range test. BI: Browning index.

**Table 4 foods-14-03420-t004:** Total phenolics, flavonoids, anthocyanins, FRAP (Ferric Reducing Antioxidant Power) and DPPH (2,2-diphenyl-1-picrylhydrazyl) antioxidant assays in blackberry jam samples.

	Total Phenolics (g GAE kg^−1^)	Total Flavonoids (g QE kg^−1^)	Total Anthocyanins (g C3GE kg^−1^)	FRAP (mmol TE kg^−1^)	DPPH (mmol TE kg^−1^)
J-F1-0P	1.96 ± 0.21 ^d^	0.87 ± 0.01 ^e^	0.09 ± 0.01 ^b^	35.73 ± 0.08 ^d^	6.74 ± 0.06 ^e^
J-F2-0P	1.08 ± 0.05 ^a–c^	0.19 ± 0.07 ^a^	0.03 ± 0.00 ^a^	25.04 ± 0.45 ^a^	5.74 ± 0.07 ^b^
J-F3-0P	1.35 ± 0.10 ^ab^	0.37 ± 0.01 ^c^	0.03 ± 0.00 ^a^	32.73 ± 1.23 ^c^	6.86 ± 0.04 ^ef^
J-F1-0.1P	1.64 ± 0.00 ^b–d^	0.71 ± 0.04 ^d^	0.12 ± 0.00 ^c^	36.08 ± 0.48 ^d^	6.76 ± 0.01 ^e^
J-F2-0.1P	1.12 ± 0.01 ^a^	0.27 ± 0.00 ^b^	0.05 ± 0.00 ^ab^	23.79 ± 1.52 ^a^	5.49 ± 0.12 ^a^
J-F3-0.1P	1.35 ± 0.02 ^ab^	0.37 ± 0.01 ^c^	0.05 ± 0.01 ^ab^	31.18 ± 0.93 ^c^	6.56 ± 0.04 ^d^
J-F1-0.5P	1.77 ± 0.00 ^cd^	0.68 ± 0.01 ^d^	0.13 ± 0.01 ^c^	36.74 ± 0.36 ^d^	6.92 ± 0.05 ^f^
J-F2-0.5P	1.38 ± 0.00 ^a–c^	0.37 ± 0.03 ^c^	0.06 ± 0.00 ^b^	31.06 ± 0.83 ^c^	6.78 ± 0.08 ^ef^
J-F3-0.5P	1.28 ± 0.01 ^ab^	0.42 ± 0.01 ^c^	0.05 ± 0.00 ^ab^	28.46 ± 0.25 ^b^	6.37 ± 0.10 ^c^

Values are expressed as mean ± standard deviation. ^a–f^ Values with different letters. in the same column are significantly different (*p* < 0.05), as determined by Duncan’s multiple range test.

**Table 5 foods-14-03420-t005:** Individual phenolic compounds (mg kg^−1^) of the blackberry jam samples.

Samples	4-aminobenzoic Acid (PABA)	Protocatechuic Acid	4-hydroxybenzoic Acid	Catechin	Chlorogenic Acid	Caffeic Acid	Epicatechin	Ferulic Acid
J-F1-0P	13.43 ± 0.40 ^d^	1.55 ± 0.08 ^e^	9.32 ± 0.21 ^b^	1.97 ± 0.14 ^f^	6.33 ± 0.14 ^b^	5.88 ± 0.31 ^f^	11.41 ± 0.36 ^g^	7.51 ± 0.25 ^c^
J-F2-0P	3.38 ± 0.29 ^a^	0.44 ± 0.08 ^b^	7.25 ± 0.17 ^a^	3.18 ± 0.12 ^g^	3.23 ± 0.13 ^a^	3.43 ± 0.17 ^d^	2.61 ± 0.21 ^b^	4.75 ± 0.09 ^b^
J-F3-0P	5.33 ± 0.35 ^b^	0.36 ± 0.05 ^a^	20.73 ± 0.37 ^f^	1.12 ± 0.10 ^c^	17.35 ± 0.36 ^d^	4.42 ± 0.15 ^e^	1.16 ± 0.08 ^a^	2.00 ± 0.11 ^a^
J-F1-0.1P	15.25 ± 0.35 ^e^	0.62 ± 0.05 ^c^	22.15 ± 0.40 ^g^	1.63 ± 0.08 ^d^	24.38 ± 0.22 ^g^	1.35 ± 0.06 ^a^	10.12 ± 0.34 ^f^	14.48 ± 0.53 ^g^
J-F2-0.1P	3.11 ± 0.27 ^a^	0.51 ± 0.03 ^b^	12.54 ± 0.40 ^d^	0.82 ± 0.05 ^b^	6.45 ± 0.27 ^b^	1.65 ± 0.08 ^b^	9.39 ± 0.23 ^e^	12.96 ± 0.40 ^f^
J-F3-0.1P	7.58 ± 0.21 ^c^	0.22 ± 0.04 ^a^	20.51 ± 0.39 ^f^	0.61 ± 0.03 ^a^	29.83 ± 0.52 ^b^	1.85 ± 0.07 ^b^	4.52 ± 0.16 ^c^	8.45 ± 0.24 ^d^
J-F1-0.5P	24.28 ± 0.68 ^f^	0.84 ± 0.07 ^e^	18.62 ± 0.41 ^e^	1.75 ± 0.06 ^e^	28.89 ± 0.24 ^a^	3.29 ± 0.08 ^d^	4.42 ± 0.27 ^c^	14.79 ± 0.41 ^g^
J-F2-0.5P	3.43 ± 0.26 ^a^	0.52 ± 0.03 ^b^	11.27 ± 0.26 ^c^	1.24 ± 0.07 ^c^	7.55 ± 0.24 ^c^	2.66 ± 0.06 ^c^	8.66 ± 0.29 ^d^	5.29 ± 0.14 ^b^
J-F3-0.5P	7.75 ± 0.45 ^c^	0.51 ± 0.06 ^b^	20.89 ± 0.26 ^f^	1.30 ± 0.04 ^c^	21.15 ± 0.26 ^e^	1.73 ± 0.04 ^b^	2.86 ± 0.06 ^b^	9.46 ± 0.25 ^e^

The values are expressed as mean ± standard deviation. ^a–g^ Values marked with different letters in the same column are significantly different (*p* < 0.05), as determined by Duncan’s multiple range test.

**Table 6 foods-14-03420-t006:** Texture profile results (TPA) of blackberry jam samples.

Samples	Gel Strength (g)	Rupture Strength (g)	Brittleness (mm)	Adhesiveness (g.s)	Consistency (g.s)
J-F1-0P	219.92 ± 13.38 ^d^	4486.65 ± 2621.92 ^b^	12.91 ± 2.95 ^c^	−5203.23 ± 550.76 ^a^	27,196.79 ± 1367.30 ^e^
J-F2-0P	69.79 ± 13.33 ^ab^	1044.30 ± 66.45 ^a^	13.70 ± 0.41 ^c^	−2631.53 ± 165.13 ^c^	7569.02 ± 357.06 ^b^
J-F3-0P	153.76 ± 73.26 ^b–d^	1375.04 ± 210.62 ^a^	11.57 ± 0.46 ^c^	−3764.64 ± 505.43 ^b^	12,229.18 ± 1584.86 ^c^
J-F1-0.1P	80.20 ± 56.19 ^a–c^	543.97 ± 365.79 ^a^	8.67 ± 4.46 ^a–c^	−526.76 ± 69.44 ^d^	7489.30 ± 1697.10 ^b^
J-F2-0.1P	80.13 ± 30.12 ^a–c^	398.71 ± 94.83 ^a^	13.22 ± 2.07 ^c^	−828.46 ± 143.25 ^cd^	3337.89 ± 354.33 ^a^
J-F3-0.1P	108.99 ± 45.50 ^a–c^	417.45 ± 81.73 ^a^	10.70 ± 2.48 ^bc^	−601.40 ± 77.21 ^d^	4317.53 ± 788.29 ^a^
J-F1-0.5P	122.24 ± 11.34 ^a–c^	294.17 ± 46.643 ^a^	4.69 ± 0.47 ^a^	−727.83 ± 28.35 ^cd^	21,109.01 ± 4674.23 ^d^
J-F2-0.5P	66.20 ± 29.79 ^a^	407.26 ± 35.53 ^a^	9.89 ± 1.91 ^a–c^	−667.66 ± 81.24 ^cd^	4291.41 ± 588.20 ^a^
J-F3-0.5P	159.68 ± 35.43 ^cd^	438.66 ± 217.84 ^a^	5.86 ± 2.34 ^ab^	−1065.74 ± 236.18 ^d^	10,378.70 ± 1979.15 ^bc^

Values are expressed as mean ± standard deviation. ^a–e^ Values with different letters in the same column are significantly different (*p* < 0.05), as determined by Duncan’s multiple range test.

**Table 7 foods-14-03420-t007:** Sensory analysis of blackberry jam samples.

Samples	Code	Odor	Color	Appearance	Taste	Consistency	Texture	Overall Acceptability
J-F1-0P	M1	2.22 ^a^	2.67 ^a^	2.33 ^ab^	1.89 ^a^	1.89 ^a^	1.33 ^a^	2.22 ^ab^
J-F2-0P	M2	2.89 ^a^	2.78 ^ab^	2.44 ^a–c^	2.44 ^a–c^	2.33 ^ab^	2.00 ^a–c^	2.44 ^a–c^
J-F3-0P	M3	2.11 ^a^	2.44 ^a^	2.00 ^a^	2.11 ^ab^	2.00 ^a^	1.67 ^ab^	2.00 ^a^
J-F1-0.1P	M4	2.44 ^a^	3.00 ^ab^	3.00 ^b–d^	2.89 ^a–d^	3.33 ^bc^	2.78 ^cd^	3.00 ^bc^
J-F2-0.1P	M5	2.56 ^a^	3.00 ^ab^	3.33 ^cd^	2.89 ^a–d^	3.22 ^bc^	3.11 ^d^	3.11 ^bc^
J-F3-0.1P	M6	2.44 ^a^	3.11 ^ab^	3.44 ^d^	3.33 ^cd^	3.56 ^c^	3.22 ^d^	3.33 ^c^
J-F1-0.5P	M7	2.89 ^a^	3.00 ^ab^	3.11 ^b–d^	2.78 ^a–d^	2.67 ^a–c^	2.56 ^b–d^	3.00 ^bc^
J-F2-0.5P	M8	2.67 ^a^	3.67 ^b^	3.22 ^b–d^	3.67 ^d^	3.33 ^bc^	3.44 ^d^	3.11 ^bc^
J-F3-0.5P	M9	2.78 ^a^	3.11 ^ab^	3.22 ^b–d^	3.22 ^b–d^	2.56 ^a–c^	2.89 ^cd^	3.11 ^bc^

Values are expressed as mean ± standard deviation. ^a–d^ Values with different letters in the same column are significantly different (*p* < 0.05), as determined by Duncan’s multiple range test.

## Data Availability

All data generated or analyzed during this study are included in this published article, further inquiries can be directed to the corresponding authors.

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
