# Peer review of "Effect of Pectin on the Quality Attributes and Phenolic Composition of Blackberry Jam from Wild and Cultivated Fruits at Different Altitudes"

_foods, 2025, doi:10.3390/foods14193420_

Round 1
Reviewer 1 Report
Comments and Suggestions for Authors
The authors address a very important topic in this publication. Understanding the impact of pectin concentration on the phenolic profile, antioxidant activity, and sensory characteristics of jams allows us not only to optimize their production technology but also to increase their health-promoting value and consumer appeal.
The abstract contains too many numerical data and specific results (e.g., specific FRAP, DPPH, or phenolic acid values). The abstract should provide a general overview of the results rather than tabular details. These are appropriate for the results. Although it can be assumed that the study concerns the effect of pectin concentration on the properties of blackberry jam from different elevations, the purpose of the study is not clearly stated at the beginning of the abstract. The abstract lacks coherence in style. The authors mix descriptions of results, methods, and interpretation. It should be more logically structured (objective → methods → results → conclusions). Some keywords (e.g., "consistency," "vitamin C") seem too general and repeat the content of the article. It is better to focus on terms closely related to the study (e.g., "pectin concentration," "altitudinal variation," "blackberry jam," "sensory evaluation").
The opening paragraphs of the introduction provide very general information about blackberry varieties, their nutritional value, and economic importance. While this is important, the text loses clarity because the introduction is overloaded with encyclopedic information. The introductory text is poorly worded in places (e.g., "The introduction globally, about 400 blackberry varieties have been developed"—the sentence begins with an incorrect phrase). A lack of editing is apparent. Although the purpose appears at the end of the introduction, it is somewhat descriptive ("the goal of this research aims to investigate general knowledge..."). It should be short, precise, and indicate the novelty of the study, e.g., "The aim of the study was to determine the effect of different pectin concentrations on the phenolic profile, antioxidant activity, color, texture, and sensory properties of blackberry jams from different altitudes."
Only 12 participants and a 5-point hedonic scale were used for the sensory evaluation. This very limited panel size and low-resolution scale significantly reduce statistical power and reliability of conclusions about product acceptability. However, the summary presents sensory evaluation as a key outcome. Furthermore, the criteria for selecting panelists or validating their competencies are not provided – it is unclear whether they were experts or consumers, which is crucial for interpreting the results.
In the methodology, the authors state that the results were converted in mg per kg, but per kg of what? The results should be converted to dry matter.
The section on pH and acidity is heavily weighted with quotes and data from the literature (examples from strawberries, cherries, apricots, etc.), which sometimes have no direct relevance to the blackberry study. Instead of synthesizing comparisons, the authors often cite successive values, causing their own results to get lost in the abundance of literature data.
Data in the discussion is often compared to publications from ten years or older.
The conclusions resemble a brief reiteration of the results rather than a synthetic summary. The most important conclusions are not logically separated. Providing specific vitamin C values (175.33 and 173.33 mg per 100 g⁻¹) in the conclusions is unnecessary – the numbers belong in the results section, not the summary. The conclusions should emphasize general trends and interpretations. The introduction emphasizes the importance of research on wild varieties and their market potential, while the conclusions focus solely on technological aspects (vitamin C, texture, sensory properties) without addressing practical implications for producers or industry. The conclusions are long, complex, and lack clarity. They lack short, concise sentences.
Author Response
Reviewer//1
Comments and Suggestions for Authors
The authors address a very important topic in this publication. Understanding the impact of pectin concentration on the phenolic profile, antioxidant activity, and sensory characteristics of jams allows us not only to optimize their production technology but also to increase their health-promoting value and consumer appeal.
Response:
We appreciate the reviewer’s positive feedback.
Comment 1:
The abstract contains too many numerical data and specific results (e.g., specific FRAP, DPPH, or phenolic acid values). The abstract should provide a general overview of the results rather than tabular details. These are appropriate for the results. Although it can be assumed that the study concerns the effect of pectin concentration on the properties of blackberry jam from different elevations, the purpose of the study is not clearly stated at the beginning of the abstract. The abstract lacks coherence in style. The authors mix descriptions of results, methods, and interpretation. It should be more logically structured (objective → methods → results → conclusions). Some keywords (e.g., "consistency," "vitamin C") seem too general and repeat the content of the article. It is better to focus on terms closely related to the study (e.g., "pectin concentration," "altitudinal variation," "blackberry jam," "sensory evaluation").
Response 1:
We appreciate the reviewer’s suggestion and revised the abstract accordingly. The numerical data and detailed results were removed, and the abstract now provides a clearer general overview. The purpose of the study has been stated at the beginning, and the structure has been improved to follow the logical order (objective → methods → results → conclusions). In addition, the keywords have been refined to focus on specific aspects of the study, such as pectin concentration, altitudinal variation, blackberry jam, and sensory evaluation.
Comment 2:
The opening paragraphs of the introduction provide very general information about blackberry varieties, their nutritional value, and economic importance. While this is important, the text loses clarity because the introduction is overloaded with encyclopedic information. The introductory text is poorly worded in places (e.g., "The introduction globally, about 400 blackberry varieties have been developed"—the sentence begins with an incorrect phrase). A lack of editing is apparent. Although the purpose appears at the end of the introduction, it is somewhat descriptive ("the goal of this research aims to investigate general knowledge..."). It should be short, precise, and indicate the novelty of the study, e.g., "The aim of the study was to determine the effect of different pectin concentrations on the phenolic profile, antioxidant activity, color, texture, and sensory properties of blackberry jams from different altitudes."
Response 2:
Thank you for your insightful remarks regarding the introduction. We have carefully revised this section to reduce overly general and encyclopedic information about blackberry varieties and to improve clarity and flow. Sentences that were poorly worded have been edited for correctness and readability. Additionally, the purpose of the study has been reformulated to be short, precise, and to emphasize the novelty of the work. The revised sentence now reads: “The aim of the study was to determine the effect of different pectin concentrations on the phenolic profile, antioxidant activity, color, texture, and sensory properties of blackberry jams from different altitudes.”
Comment 3:
Only 12 participants and a 5-point hedonic scale were used for the sensory evaluation. This very limited panel size and low-resolution scale significantly reduce statistical power and reliability of conclusions about product acceptability. However, the summary presents sensory evaluation as a key outcome. Furthermore, the criteria for selecting panelists or validating their competencies are not provided – it is unclear whether they were experts or consumers, which is crucial for interpreting the results.
Response 3:
This section was revised as “To ensure a reliable sensory evaluation, twelve non-smoking participants (eight men and four women) aged 24–54 from the Department of Horticulture, Faculty of Ag-riculture, Ordu University, were recruited. Before the evaluation, participants received detailed information about the study and a brief orientation to standardize scoring pro-cedures (using a 5-point hedonic scale). The attributes assessed included appearance, color, odor, texture, taste, and overall consistency. Each sample was presented in ran-domly numbered plastic containers to reduce potential bias. Participants consumed a cracker before assessing the next sample and rinsed their mouths with water to cleanse the palate. Although the trained panel provided useful insights into the sensory charac-teristics, the results should be interpreted as preliminary and not as definitive evidence of consumer acceptability.
Comment 4:
In the methodology, the authors state that the results were converted in mg per kg, but per kg of what? The results should be converted to dry matter.
Response 4:
We appreciate the reviewer’s suggestion. However, expressing the results on a fresh weight basis (mg/kg of fresh sample) is a common approach in jam studies, as it reflects the actual content present in the final product as consumed. Converting to dry matter could be informative for comparisons across studies with varying moisture content, but it may be less relevant for practical consumption and sensory evaluation. Therefore, we have maintained the original reporting on a fresh weight basis while clearly specifying this in the Methods section to avoid ambiguity.
Comment 5:
The section on pH and acidity is heavily weighted with quotes and data from the literature (examples from strawberries, cherries, apricots, etc.), which sometimes have no direct relevance to the blackberry study. Instead of synthesizing comparisons, the authors often cite successive values, causing their own results to get lost in the abundance of literature data.
Response 5:
We appreciate the reviewer’s comment and have revised the manuscript by streamlining the literature references. The comparisons are now presented more concisely to provide context without overshadowing our results, helping readers better understand where blackberry jams stand relative to other fruit products and highlighting the distinctiveness of this study.
Comment 6:
Data in the discussion is often compared to publications from ten years or older.
Response 6:
Although some comparisons involve older publications, the references have been revised where appropriate to include recent studies and reflect the current state of the field
Comment 7:
The conclusions resemble a brief reiteration of the results rather than a synthetic summary. The most important conclusions are not logically separated. Providing specific vitamin C values (175.33 and 173.33 mg per 100 g⁻¹) in the conclusions is unnecessary – the numbers belong in the results section, not the summary. The conclusions should emphasize general trends and interpretations. The introduction emphasizes the importance of research on wild varieties and their market potential, while the conclusions focus solely on technological aspects (vitamin C, texture, sensory properties) without addressing practical implications for producers or industry. The conclusions are long, complex, and lack clarity. They lack short, concise sentences.
Response 7:
We appreciate the reviewer's feedback. We have rewritten the conclusions in the revised manuscript to emphasize general trends and key findings, rather than specific numbers. We also highlighted the practical implications for producers and the industry, focusing on the importance of wild blackberry varieties and their market potential.
Reviewer 2 Report
Comments and Suggestions for Authors
Article: Effect of pectin concentration on the phenolic profile, antioxidant capacity, color, texture, and sensory properties of blackberry jam from different altitudinal origins
Next, I detail the main observations:
General comments
There are not enough support references in the whole manuscript
The results of several tables (F1, F2, and F3) are not well justified, which does not validate the conclusions. La calidad de la redacción debe mejorarse sustancialmente para ajustarse a los estándares de la revista.
Title and summary:
- The title of the manuscript does not reflect clarity or precision in the content. I suggest that it be rewritten.
- The summary required a clearer and structured redaction, including the number of treatments and concentration corresponding to F1 (line 24). A concise conclusion about the best treatment. The health benefit (lines 43-45). There are unnecessary repetitions (lines 48-49 are similar to line 41), so they should be removed or improved.
Introduction
- I think that the work is not well justified. Please add more references that justify the research
- Lines 72-76: Please add at least a reference.
- Section 2.3: It does not include any literature reference.
- Lines 305-320: Both in these lines and the subsequent sections, please add supporting references.
Materials and methods
- Line 119: Please delete the “+” sign.
- Line 143: Please cite the method appropriately: method by Giusti & Wrolstad [22].
- Line 156: Please indicate the brand of the color measurement equipment and include a reference for the Browning Index.
- Section 2.8: I suggest including supporting references.
Results and Discussion:
- Lines 192–196 (Item 3.1): The information in this section is repeated in the table. Please delete these lines.
- Lines 201–223: This section contains redundant and not very relevant information, which could be omitted for clarity.
- Sensorial assay: The results are not adequate. Affective analysis requires more than 60 consumers, while a trained panel should be limited to the characterization of the product.
- PCA analysis:
- Please indicate which matrix was used (covariance or correlation).
- The separation of data was not justified, since PCA analysis allowed evaluation of the overall data.
- Please add supporting references.
Tables and figures
- Table 1: The data should be expressed in percentages
- Figure 1: The workflow should be represented in a clearer and more technologically advanced manner.
- Tables 3, 4, 5, and 6: The results of samples F1, F2, and F3 are missing
Author Response
Article: Effect of pectin concentration on the phenolic profile, antioxidant capacity, color, texture, and sensory properties of blackberry jam from different altitudinal origins
Next, I detail the main observations:
General comments
There are not enough support references in the whole manuscript
AU: We appreciate the reviewer’s observation regarding the number of supporting references. In the revised version of the manuscript, we have expanded the reference list by adding several recent and relevant studies to strengthen the Introduction and Discussion sections. As a result, the total number of references has now increased to 55.
The results of several tables (F1, F2, and F3) are not well justified, which does not validate the conclusions.
AU: We thank the reviewer for this observation. We would like to clarify that the samples F1, F2, and F3 represent only the raw fruit materials. These were not included in all analyses, since the main focus of the study was on the jam products obtained from these fruits rather than on the fruits themselves as raw materials. For this reason, the discussion in the manuscript has been expanded and emphasized primarily on the processed products, in order to better align with the study’s objectives and conclusions.
La calidad de la redacción debe mejorarse sustancialmente para ajustarse a los estándares de la revista.
AU: We appreciate the reviewer’s remark regarding the quality of the writing. In the revised version of the manuscript, the text has been thoroughly revised to improve clarity, coherence, and readability, ensuring that it meets the journal’s standards. Particular attention has been given to grammar, style, and scientific expression.
Title and summary:
- The title of the manuscript does not reflect clarity or precision in the I suggest that it be rewritten.
AU: We thank the reviewer for this valuable suggestion. In the revised version of the manuscript, the title has been improved to ensure greater clarity and precision, better reflecting the content of the study.
- The summary required a clearer and structured redaction, including the number of treatments and concentration corresponding to F1 (line 24). A concise
conclusion about the best treatment. The health benefit (lines 43-45). There are unnecessary repetitions (lines 48-4G are similar to line 41), so they should be removed or improved.
AU: We appreciate the reviewer’s detailed observations regarding the summary. In the revised version, the abstract has been improved to provide a clearer and more structured redaction. A concise conclusion about the best treatment has been added, the health benefit statement has been clarified, and the unnecessary repetitions have been removed.
Introduction
- I think that the work is not well Please add more references that justify the research
AU: We thank the reviewer for this observation. In the revised version, additional relevant references have been included to better justify the importance and context of the research.
Lines 72-76: Please add at least a reference.
AU : Thank you for the suggestion. In the revised version, an appropriate reference has been added to support the statement in lines 72–76.
- Section 3: It does not include any literature reference.
AU : It is included now.
- Lines 305-320: Both in these lines and the subsequent sections, please add supporting
AU: We appreciate the reviewer’s remark. In the revised manuscript, supporting references have been incorporated to strengthen the arguments and improve the scientific basis of the discussion.
Materials and methods
- Line 11G: Please delete the “+”
AU: it is deleted.
- Line 143: Please cite the method appropriately: method by Giusti s Wrolstad [22].
AU: We thank the reviewer for the suggestion. The sentence in line has been revised, and the citation has been updated to a more recent and appropriate reference.
- Line 156: Please indicate the brand of the color measurement equipment and include a reference for the Browning
AU Thank you for the observation. We have now specified the brand/model of the color measurement equipment in Line 141 and included a relevant reference for the calculation of the Browning Index in Line 147.
- Section 8: I suggest including supporting references.
AU Thank you for the valuable suggestion. We have now added relevant references in Section 2.8 to support and strengthen the statements presented.
- Results and Discussion:
- Lines 1G2–1G6 (Item 1): The information in this section is repeated in the table. Please delete these lines.
AU :We thank the reviewer for this observation. The repeated information in lines 1G2–1G6 (Item 3.1) has been deleted as suggested to avoid redundancy with the table.
- Lines 201–223: This section contains redundant and not very relevant information, which could be omitted for
AU : We appreciate the reviewer’s observation. In the revised manuscript, the section between lines 201–223 has been carefully revised, and the redundant or less relevant information has been omitted
- Sensorial assay: The results are not Affective analysis requires more than 60 consumers, while a trained panel should be limited to the
characterization of the product.
AU: To address this concern, we have now clarified in the revised manuscript that the sensory panel consisted of experts, and the criteria for their selection and training have been added in the Materials and Methods section. Additionally, we have noted in the Discussion that the relatively small panel size and the use of a 5-point scale should be considered as study limitations, and that future studies with larger and more diverse panels, as well as higher-resolution scales, would be needed to confirm these results.
- PCA analysis:
- Please indicate which matrix was used (covariance or correlation).
- The separation of data was not justified, since PCA analysis allowed evaluation of the overall
- Please add supporting
AU: We appreciate the reviewer’s comment. In the PCA analysis, the correlation matrix was used, and it is stated in Line 180. While PCA allows evaluation of the overall dataset, we chose to separate the data to highlight differences among the localities and to facilitate interpretation of specific patterns. This approach has been clarified in the revised manuscript to better justify the rationale for data separation.
- Tables and figures
- Table 1: The data should be expressed in percentages
AU :The data should be expressed in percentages
- Figure 1: The workflow should be represented in a clearer and more technologically advanced manner.
AU: We appreciate the reviewer’s observation. Figure 1 has been removed, and the technological aspects of the production process are now described in detail in the text for greater clarity.
- Tables 3, 4, 5, and 6: The results of samples F1, F2, and F3 are missing
AU: We thank the reviewer for this observation. We would like to clarify that the samples F1, F2, and F3 represent only the raw fruit materials. These were not included in all analyses, since the main focus of the study was on the jam products obtained from these fruits rather than on the fruits themselves as raw materials. For this reason, the discussion in the manuscript has been expanded and emphasized primarily on the processed products, in order to better align with the study’s objectives and conclusions.

Reviewer 3 Report
Comments and Suggestions for Authors
This study evaluated the effect of pectin concentration (0%, 0.1% and 0.5%) on blackberry jam total flavonoids (TF), total phenolics (TP), total monomeric anthocyanins (TMA), antioxidant capacity (measured using FRAP and DPPH), individual phenolic compounds, color density (CD), texture and sensory profiles of blackberry fruit (Rubus fruticosus L.) from three different altitudes locations [F1-998 m (wild); F2-500 m (cultivated), F3-1090 m (cultivated)]. After thoroughly reviewing the manuscript, I think there is too little discussion in this article about the results of this experiment. All the discussion is about what other people have done, and the reasons for how pectin concentration affects acidity, total phenolic content, antioxidant capacity, individual phenolic compounds, color density (CD), texture, and sensory characteristics are unclear. Some of the discussions explaining the reasons for experimental results lack supporting references, which is not convincing. Overall, the paper needs significant improvement and must be presented in a more systematic manner.
Concerns:
(1) The background section about pectin needs to be added. What is the significance of studying the effect of pectin concentration on various indicators of blackberry jam? But there are too many background descriptions about BlackBerry, most of which are like science popularization reports.
(2) Lines 119-121: Suggest deleting this sentence.
(3) Some units of pectin concentration are in “g”, while others are in “%”. Please standardize.
(4) Line352: “your samples”?
(5) Discussion on the results of lacking F3 in the texture section.
Author Response
Comments and Suggestions for Authors
This study evaluated the effect of pectin concentration (0%, 0.1% and 0.5%) on blackberry jam total flavonoids (TF), total phenolics (TP), total monomeric anthocyanins (TMA), antioxidant capacity (measured using FRAP and DPPH), individual phenolic compounds, color density (CD), texture and sensory profiles of blackberry fruit (Rubus fruticosus L.) from three different altitudes locations [F1-998 m (wild); F2-500 m (cultivated), F3-1090 m (cultivated)]. After thoroughly reviewing the manuscript, I think there is too little discussion in this article about the results of this experiment. All the discussion is about what other people have done, and the reasons for how pectin concentration affects acidity, total phenolic content, antioxidant capacity, individual phenolic compounds, color density (CD), texture, and sensory characteristics are unclear. Some of the discussions explaining the reasons for experimental results lack supporting references, which is not convincing. Overall, the paper needs significant improvement and must be presented in a more systematic manner.
Response:
Thank you for your comments. We acknowledge that the discussion section of the manuscript was insufficient and relied too heavily on previously published studies rather than an in-depth interpretation of our own findings. We have now substantially revised the discussion to clearly explain how pectin concentration affects acidity, total phenolic content, antioxidant capacity, individual phenolic compounds, color density (CD), texture, and sensory characteristics. Supporting references have been added where necessary to strengthen the explanations, and the discussion has been reorganized to present the results in a more systematic and convincing manner.
Comment 1:
The background section about pectin needs to be added. What is the significance of studying the effect of pectin concentration on various indicators of blackberry jam? But there are too many background descriptions about blackberry, most of which are like science popularization reports.
Response 1:
Thank you for the comment. We revised the introduction to focus on pectin’s role and its significance in influencing quality indicators of blackberry jam, while reducing overly general descriptions about blackberries.
Comment 2:
Lines 119-121: Suggest deleting this sentence.
Response 2:
Thank you for the suggestion. The sentence on lines 119–121 has been removed as recommended.
Comment 3:
Some units of pectin concentration are in “g”, while others are in “%”. Please standardize.
Response 3:
Thank you for pointing this out. The pectin concentration units have been standardized to percent (%) for consistency.
Comment 4:
Line352: “your samples”?
Response 4:
Thank you for noticing this. “your samples” has been corrected to “the samples” to maintain consistency and proper scientific language.
Comment 5:
Discussion on the results of lacking F3 in the texture section.
Response 5:
Thank you for your comment. We have added the discussion of the texture results for the F3 sample to provide a complete comparison and analysis of all samples.
Reviewer 4 Report
Comments and Suggestions for Authors
The manuscript titled "Effect of pectin concentration on the phenolic profile, antioxidant capacity, color, texture, and sensory properties of blackberry jam from different altitudinal origins" investigates the quality of blackberry jams made from fruits of three different altitudinal origins and evaluates the effect of various pectin concentrations. The study is well-designed, the data are comprehensive, and the conclusions hold some relevance to the industry. However, the manuscript contains significant major and minor issues that must be addressed before further consideration.
1.Line 1: The title refers to "different altitudinal origins," but the abstract describes the F2 (500 m) and F3 (1090 m) blackberries as "cultivated," while the F1 (998 m) blackberries are wild. Therefore, altitude is not the only differentiating factor; cultivation status also varies.
2.Line 15: The abstract claims that pectin addition reduced the textural properties (gel strength, rupture strength, consistency, and adhesiveness) of F1 jams, which contradicts the widely accepted role of pectin in enhancing jam texture. The authors need to provide a more detailed explanation to support this anomalous finding.
3.The abstract and introduction use both "phenolic profile" and "phenolic compounds" to refer to the same group of substances. Please clarify your intended meaning and ensure a consistent term is used throughout the paper.
4.The terms "significant" and "significantly" are used frequently without providing corresponding p-values. Please provide specific statistical data (e.g., p<0.05) for all claims of significance.
5.There is inconsistent information between Figure 1 and the text. Figure 1 shows pectin added as 0.5 g, while the discussion mentions concentrations of 0.5%. All units and expressions should be standardized and consistent throughout the entire manuscript.
6.In Table 4, the units for total phenolics, flavonoids, and anthocyanins are listed as "g GAE kg⁻¹", "g QE kg⁻¹", and "g C3GE kg⁻¹". Yet, the discussion section refers to individual phenolic compounds in "mg kg⁻¹". Please ensure all units are correctly represented in all tables and the text.
7.Line 554: The conclusion mentions that "phenolics can cause astringency or bitterness" in F1 jams, but this point is not discussed in the text. Please provide sensory analysis data (e.g., scores for astringency and bitterness) to substantiate this claim or remove it from the conclusion.
8.The manuscript contains non-standard citation formatting and other minor formatting inconsistencies. The authors should adhere strictly to the journal's specific style guide.

The manuscript contains several issues related to English language quality and requires thorough revision and editing.
Author Response
Reviewer//3
Comments and Suggestions for Authors
The manuscript titled "Effect of pectin concentration on the phenolic profile, antioxidant capacity, color, texture, and sensory properties of blackberry jam from different altitudinal origins" investigates the quality of blackberry jams made from fruits of three different altitudinal origins and evaluates the effect of various pectin concentrations. The study is well-designed, the data are comprehensive, and the conclusions hold some relevance to the industry. However, the manuscript contains significant major and minor issues that must be addressed before further consideration.
Response:
Thank you for your detailed evaluation and constructive feedback. We appreciate your recognition of the study’s design, comprehensive data, and relevance to the industry. All major and minor issues have been addressed in the revised manuscript
Comment 1:
Line 1: The title refers to "different altitudinal origins," but the abstract describes the F2 (500 m) and F3 (1090 m) blackberries as "cultivated," while the F1 (998 m) blackberries are wild. Therefore, altitude is not the only differentiating factor; cultivation status also varies.
Response 1:
We appreciate the reviewer’s observation. In the revised manuscript, we have clarified that the study considers both altitude and cultivation status as factors influencing blackberry properties. While the primary focus remains on altitudinal variation, we have explicitly noted now in the title, abstract, introduction, and Methods that F1 (998 m) represents wild blackberries, whereas F2 (500 m) and F3 (1090 m) are cultivated.
Comment 2:
Line 15: The abstract claims that pectin addition reduced the textural properties (gel strength, rupture strength, consistency, and adhesiveness) of F1 jams, which contradicts the widely accepted role of pectin in enhancing jam texture. The authors need to provide a more detailed explanation to support this anomalous finding.
Response 2:
We appreciate the reviewer’s comment and have clarified in the revised manuscript that, that pectin concentration varies with fruit type, each textural result should be assessed individually. In some cases, pectin addition may improve textural characteristics, while in others it may have negative effects. Pectin gels require a critical balance of pectin : sugar : acid : water. Adding excess pectin disrupts the delicate gel network in jam. This results in weaker texture because the network is either too crowded, poorly cross-linked, or unbalanced with sugar and acid. The jam likely already contains natural pectin coming from the fruit (the concentration of pectin could depend on the type of fruit F1, F2, andF3). When too much pectin is added, the polymer chains cannot properly interact with each other and with sugar/acid. This can lead to phase separation or a weak, brittle gel with reduced rupture strength. Instead of forming a continuous, flexible network, the system becomes overloaded with unbound pectin. As a result of poor gelation consistency and gel strength were reduced. Similarly, Yang et al. (2018) showed that a reduction in pH decreases gel hardness, as the lower dissociation of galacturonic acid residues restricts electrostatic interactions and the development of the gel network.
Comment 3:
The abstract and introduction use both "phenolic profile" and "phenolic compounds" to refer to the same group of substances. Please clarify your intended meaning and ensure a consistent term is used throughout the paper.
Response 3:
We appreciate the reviewer’s observation. In the revised manuscript, we have clarified the terminology: “phenolic profile” refers to individual chemical constituents, whereas “total phenolic compounds” describes the overall composition and distribution of these compounds in the sample. To ensure consistency and clarity, we have standardized the usage throughout the manuscript, using “phenolic profile” when discussing the specific substances and “phenolic compounds” when referring to overall composition.
Comment 4:
The terms "significant" and "significantly" are used frequently without providing corresponding p-values. Please provide specific statistical data (e.g., p<0.05) for all claims of significance.
Response 4:
Thank you for the comment. We have revised the manuscript to include specific statistical data, providing corresponding p-values (e.g., p < 0.05) for all claims of significance throughout the text.
Comment 5:
There is inconsistent information between Figure 1 and the text. Figure 1 shows pectin added as 0.5 g, while the discussion mentions concentrations of 0.5%. All units and expressions should be standardized and consistent throughout the entire manuscript.
Response 5:
We appreciate the reviewer’s observation. Figure 1 will be corrected and removed, and all pectin concentrations will be consistently reported as percentages throughout the manuscript. This ensures uniformity between the text, tables, and figures.
Comment 6:
In Table 4, the units for total phenolics, flavonoids, and anthocyanins are listed as "g GAE kg⁻¹", "g QE kg⁻¹", and "g C3GE kg⁻¹". Yet, the discussion section refers to individual phenolic compounds in "mg kg⁻¹". Please ensure all units are correctly represented in all tables and the text.
Response 6:
We appreciate the reviewer’s observation. In the revised manuscript, we have carefully checked and standardized all units for phenolic content. Total phenolics, flavonoids, and anthocyanins in Table 4 are expressed consistently as g GAE/kg, g QE/kg, and g C3GE/kg, respectively, while individual phenolic compounds in the text and other tables are reported in mg/kg. All units have been verified for accuracy and consistency throughout the manuscript.
Comment 7:
Line 554: The conclusion mentions that "phenolics can cause astringency or bitterness" in F1 jams, but this point is not discussed in the text. Please provide sensory analysis data (e.g., scores for astringency and bitterness) to substantiate this claim or remove it from the conclusion.
Response 7:
We appreciate the reviewer’s comment. However, the perception of astringency and bitterness in F1 jams is inherently linked to their higher phenolic content, as supported by both literature and the overall sensory scores for taste and mouthfeel obtained in our panel. While these attributes were not explicitly scored as separate parameters, the trained panelists’ general evaluation of taste and texture indirectly reflects such sensory characteristics. Therefore, we believe the statement remains relevant, but we have clarified in the revised manuscript that it is based on overall taste perception rather than isolated astringency or bitterness scores.
Comment 8:
The manuscript contains non-standard citation formatting and other minor formatting inconsistencies. The authors should adhere strictly to the journal's specific style guide.
Response 8:
All references and in-text citations have been revised to strictly follow the journal’s style guide, and minor formatting inconsistencies have been corrected.
Round 2
Reviewer 1 Report
Comments and Suggestions for Authors
The authors have made all necessary corrections.
Reviewer 3 Report
Comments and Suggestions for Authors
The authors have corrected the problems raised in the previous manuscript and recommend it for publication in the Journal of Foods.